

# Effect of interaction between occupational stress and polymorphisms of MTHFR gene and SELE gene on hypertension

Fen Yang[1], Ruiying Qiu[1], Saimaitikari Abudoubari[1], Ning Tao[1,3] and Hengqing An[2,3]

[1] School of Public Health, Xinjiang Medical University, Urumqi Xinjiang, China
[2] The First Affiliated Hospital, Xinjiang Medical University, Urumqi Xinjiang, China
[3] Xinjiang Clinical Research Center for Genitourinary System, Urumqi Xinjiang, China

Corresponding authors
Ning Tao, 38518412@qq.com
Hengqing An,
13201226586@163.com

## ABSTRACT

**Background:** Gene-environment interaction is related to the prevalence of hypertension, but the impact of genetic polymorphisms on hypertension may vary due to different geography and population.

**Objective:** To explore the impact of the interaction among occupational stress and MTHFR gene and SELE gene polymorphism on the prevalence of hypertension in Xinjiang oil workers.

**Methods:** A case-control study was conducted on 310 oil workers. In an oilfield base in Karamay City, Xinjiang, 155 hypertensive patients aged 18~60 years old with more than one year of service were selected as the case group, and 155 oil workers without hypertension were selected as the control group according to the 1:1 matching principle (matching conditions: the gender and shift were the same. The age is around 2 years old). The Occupational Stress Scale was used to evaluate the degree of occupational stress, PCR technique was used to detect MTHFR and SELE gene polymorphism, Logistic regression analysis was used to analyze the effects of gene and occupational stress on hypertension, and gene-gene and gene-environment interactions were analyzed by generalized multi-factor dimension reduction method.

**Results:** The G98T polymorphism of SELE gene ($\chi^2 = 6.776$, $P = 0.034$), the C677T ($\chi^2 = 7.130$, $P = 0.028$) and A1298C ($\chi^2 = 12.036$, $P = 0.002$) loci of MTHFR gene and the degree of occupational stress ($\chi^2 = 11.921$, $P = 0.003$) were significantly different between the case group and the control group. The genotypes GT at the G98T polymorphism of the SELE gene ($OR = 2.151$, 95% CI [1.227–3.375]), and the dominant model (AC/CC vs AA, $OR = 1.925$, 95% CI [1.613–3.816]); AC and CC at the A1298C polymorphism of the MTHFR gene ($OR_{AC} = 1.917$, 95% CI [1.064–3.453]; $OR_{CC} = 2.233$, 95% CI [1.082–4.609]), the additive model (CC vs AA, OR = 2.497, 95% CI [1.277–4.883]) and the dominant model (AC/CC vs AA, OR = 2.012, 95% CI [1.200–3.373]); at the C677T polymorphism of the MTHFR gene CT and TT ($OR_{CT} = 1.913$, 95% CI [1.085–3.375]; $OR_{TT} = 3.117$, 95% CI [1.430–6.795]), the additive model (CC vs AA, OR = 1.913, 95% CI [1.085–3.375]) and the dominant model (AC/CC vs AA, OR = 2.012, 95% CI [1.200–3.373]), which could increase hypertension risk ($P < 0.05$). The gene-gene interaction showed that there was a positive interaction between the A1298C and C677T sites of the MTHFR gene, and the gene-occupational stress interaction showed that there was a

positive interaction between the A1298C and C677T sites of the MTHFR gene and the occupational stress.

**Conclusion:** The interaction of gene mutation and occupational stress in Xinjiang oil workers maybe increase the risk of hypertension.

## INTRODUCTION

High systolic blood pressure is a leading cause of death from ischemic heart disease, found in Global Burden of Disease Study 2019 (*GBD 2019 Risk Factors Collaborators, 2019*). Therefore, the effective prevention and control of hypertension in the population has become a major public health issue. Data from a cardiovascular disease risk screening study of 170,000 urban and rural residents showed that the detection rate of age-specific hypertension in people aged 35–75 was as high as 37% (*Lu et al., 2017*). However, the etiology of hypertension is complex and diverse, and has not yet been comprehensively studied. It has been shown that genetic and environmental interactions result in an increased risk of developing hypertension (*Huang et al., 2019*; *Kokubo et al., 2019*). Heritability studies have confirmed that genetic variation among individuals with hypertension accounts for approximately 30–60% of the variation (*Liu et al., 2015*). In recent years, a number of investigators have focused on the association of *MTHFR* gene (*McNulty et al., 2017*) *SELE* gene (*Liao et al., 2016*) polymorphisms with hypertension, but the findings have been inconsistent. Evidence from both the GWAS (*Lu et al., 2015*) and epidemiological studies (*Amenyah et al., 2020*; *Ji et al., 2019*) suggests that MTHFR gene polymorphisms are associated with the risk of developing hypertension, but polymorphisms at the C677T and A1298C loci of the MTHFR gene are not consistent in all populations. A case-control study in Orléans found that the prevalence of hypertension in the Algerian population was not associated with the MTHFR gene (*Amrani-Midoun et al., 2016*). Many studies have also confirmed that subjects carrying the 677CC genotype are at significantly increased risk of developing hypertension, while the C677T gene polymorphism leads to lower blood pressure (*Ghogomu et al., 2016*). Meta-analysis suggests that carriers of the C allele of the A561C polymorphism of the SELE gene may contribute to an increased risk of hypertension in the Chinese Han population (*Ouyang et al., 2015*). In addition, a significant association between G98T polymorphism and hypertension has also been found (*Chen et al., 2008*). From this it is clear that the probability of developing cardiovascular risk varies geographically due to genetic polymorphisms. Oil workers in Karamay, Xinjiang, are often associated with the occurrence of occupational stress due to the nature of their special occupation, which in turn affects their physical and mental health. A number of studies have now shown that occupational stress may cause an increase in blood pressure through the neurological response of individuals (*Tao et al., 2018*).

Therefore, considering the variation in the results of studies conducted in different populations, this study will comprehensively investigate the effects of the interaction among MTHFR gene and SELE gene polymorphisms and occupational stress on hypertension, that aim to provide new ideas for the prevention and control of hypertension in oil workers.

# MATERIALS AND METHODS

## Study subjects

In the case-control study, a total of 183 oil workers with hypertension according to the results of physical examination in 2020 at the central hospital in Karamay, excluding 28 people with incomplete information (missing questionnaire information, no blood samples, low DNA extraction concentration, etc.), and finally including 155 hypertensive patients aged between 18–60 years with 1 year of work experience as the case group, and 155 non-hypertensive patients were selected as the control group according to the 1:1 matching principle with gender, age (±2 years), and shift status as matching factors. All participants signed a written informed consent. The study was approved by the Ethics Review Committee of the First Affiliated Hospital of Xinjiang Medical University (No. 2015006).

# METHODS

## General information and occupational stress survey

A structured questionnaire was used to collect information on gender, age, educational level, shift work, smoking and alcohol consumption (Smoker: smoking ≥1 cigarette per day for six months or more; drinker: drinking ≥2 times a week with alcohol intake ≥50 g per drinking session regularly ≥1 year (*Xing et al., 2018*)), marital status, working age, personal income per month and BMI among oil workers. Among them, the Occupational Stress Inventory Revised Edition (OSI-R) was used to assess the level of occupational stress. The scale consists of the Occupational Role Questionnaire (ORQ), the Personal Strain Questionnaire (PSQ), and the Personal Resources Questionnaire (PRQ).

The occupational stress of oil workers in this study was assessed by the ORQ, which consists of six dimensions with 10 entries in one dimension and using a scale of 1 to 5 and summarized, with higher ORQ scores representing higher levels of occupational stress. According to the scoring principle of the scale, occupational stress was classified as: "Low" (ORQ > 160), "Middle" (120 ≤ ORQ ≤ 160), and "High" (ORQ < 120).

## Diagnosis of hypertension and diagnostic criteria

Blood pressure was measured by a professional physician at the Karamay Central Hospital during a routine physical examination of the oil workers. After the oil workers rested for 10 min, blood pressure was measured twice at 5 min intervals using a zeroed standard mercury sphygmomanometer to take the mean value, and when the difference was greater than 5 mmHg, the mean value was taken after the third measurement. In this study, hypertension was defined according to the Chinese Guidelines for the Prevention and Treatment of Hypertension (2018 revision) (*Liu, 2019*) as (1) systolic blood pressure (SBP)

**Table 1** The primers and restriction endonuclease for amplifying the PCR of the target SNPs.

| | SELE A 561C | SELE G98T | MTHFR C677T | MTHFR A1298C |
|---|---|---|---|---|
| Primer sequences | F:ATTAGCATCAAGGTTTAGGATAGGT R:TGAAGAAAGAGAGGCAAGAACCA | F:TGCCCAAAATCTTAGGATGC R:AAGCCCAGGGAAGAACACAT | F:TGAAGGAGAGGTGTCTGCGGGA R:AGGACGGTGCGGTGAGAGTG | F:CTTCTACCTGAAGAGCAAGTC R:CATGTCCACAGCATGGAG |
| Length | 326 bp | 332 bp | 198 bp | 256 bp |
| REE (µl) | PstI(0.2) | HphI(1) | HinfI(1) | MboII(1) |
| 10x Buffer (µl) | 2 | 2 | 2 | 2 |
| ddH2O (µl) | 3 | 8.5 | 7 | 7 |
| PCR products | 17 | 8.5 | 10 | 10 |
| Temperature (°C) | 37 | 37 | 37 | 37 |
| Time (min) | 15 | 240 | 10 | 60 |
| Agarose gels | 2.5% | 2.5% | 3% | 3% |
| Voltage | 110 V | 100 V | 110 V | 100 V |
| Genotyping | CC:326 bp | TT:332 bp | CC:198 bp | CC256 bp |
| | AA:222+104 bp | GG:194+138 bp | TT:175+23 bp | AA:17+22+30+27 bp |
| | AC:326+222+104 bp | GT:332+194+138 bp | CT:198+175+23 bp | AC:256+176+22+30+27 bp |

≥140 mmHg and/or diastolic blood pressure (DBP) ≥90 mmHg and (2) self-reported hypertension diagnosed by a physician, and being treated with antihypertensive therapy within the past two weeks.

## Sample collection

A total of 4 ml venous blood sample was collected from each subject after 8 h fasting period. The samples were put into the ethylenediaminetetraacetic acid (EDTA) anticoagulation tubes. The serum and plasma were separated by centrifugation at 4 °C, 8 000 r/min (20 mm radius), for 3 min, and then stored at −20 °C for future and used to extract the genomic deoxyribonucleic acids (DNAs) and ribonucleic acids (RNAs).

## Genotyping method of MTHFR and SELE gene polymorphisms

DNA in blood was quantified using spectrophotometric analysis, and a polymerase chain reaction-restriction fragment length polymorphism (PCR-RELP) technique to amplify the target fragment and digest the PCR amplified product with specific endonucleases for SNP typing, As follow:

First, the genomic DNA was extracted using a TIANamp Blood DNA Kit (TIANGEN, China) in accordance with the manufacturer's instructions. Microspectrophotometer was used to detect the concentration and purity of the DNA samples for the subsequent experiments. Concentration >100 µg/µl, 1.70 <260OD/280OD <2.0 means that the sample is qualified. Second, all DNA samples were genotyped through polymerase chain reaction (PCR)–ligase detection reaction. Table 1 presents the primers and restriction endonuclease for amplifying the PCR of the target SNPs for each participant.

## Statistical analysis

Comparison of rates and genotypes between groups for each polymorphism were performed using chi-square tests, all in SPSS 25.0. SHEsis software was used for

Hardy-Weinberg equilibrium test. The effect of each gene polymorphism and occupational tension on hypertension was analyzed by logistic regression. Gene-gene and gene-occupational tension interactions were modeled by applying generalized multifactor downscaling (GMDR) software, and dendrograms were drawn with Multifactor dimensionality reduction (MDR) method.

## RESULTS

### Distribution of basic demographic characteristics of the case and control groups

In this study, 310 oil workers (155 in the case group and 155 in the control group) were investigated, of whom 200 were male and 110 were female in Table 2. The comparison of hypertension among demographic characteristics showed that the differences between different personal income per month ($\chi^2$ = 5.684, $P$ = 0.017), smoking ($\chi^2$ = 27.01, $P < 0.001$), drinking alcohol ($\chi^2$ = 8.903, $P$ = 0.003), and BMI ($\chi^2$ = 5.845, $P$ = 0.016) were statistically significant, while the differences between different ethnic group, educational level, professional title, marital status, and working age groups were not statistically significant ($P > 0.05$).

### Distribution of SELE and MTHFR genotypic loci and occupational stress between the case group and control group

The distribution of SELE and MTHFR genotype loci and occupational stress degree between the case group and the control group is shown in Table 3. Hardy-Weinberg genetic balance test showed that the distribution of G98T and A561C loci of SELE gene, A1298C and C677T loci of MTHFR gene were consistent with expected values in case group and control group. The G98T polymorphism of SELE gene ($\chi^2$ = 6.776, $P$ = 0.034), the C677T ($\chi^2$ = 7.130, $P$ = 0.028) and A1298C ($\chi^2$ = 12.036, $P$ = 0.002) loci of MTHFR gene and the degree of occupational stress ($\chi^2$ = 11.921, $P$ = 0.003) were significantly different between the case group and the control group.

### Logistic regression analysis of the relationship between genetic and occupational stress with hypertension

The association of SELE and MTHFR genes and occupational stress with hypertension was analyzed using binary logistic regression and is shown in Table 4. After adjusting for personal income per month, smoking, drinking alcohol, and BMI, genotypes GT at the G98T polymorphism of the SELE gene (OR = 2.151, 95% CI [1.227–3.375]), and the dominant model (AC/CC $vs$ AA, OR = 1.925, 95% CI [1.613–3.816]); AC and CC at the A1298C polymorphism of the MTHFR gene ($OR_{AC}$ = 1.917, 95% CI [1.064–3.453]; $OR_{CC}$ = 2.233, 95% CI [1.082–4.609]), the additive model (CC $vs$ AA, OR = 2.497, 95% CI [1.277–4.883]) and the dominant model(AC/CC $vs$ AA, OR = 2.012, 95% CI [1.200–3.373]); at the C677T polymorphism of the MTHFR gene CT and TT ($OR_{CT}$ = 1.913, 95% CI [1.085–3.375]; $OR_{TT}$ = 3.117, 95% CI [1.430–6.795]), the

**Table 2 Distribution of basic demographic characteristics of case group and control group.**

| Item | N | Case group | | Control group | | $\chi^2$ | P |
|---|---|---|---|---|---|---|---|
| | | n | % | n | % | | |
| Gender | | | | | | 0.000 | 1.000 |
| Male | 200 | 100 | 50.0 | 100 | 50.0 | | |
| Female | 110 | 55 | 50.0 | 55 | 50.0 | | |
| Age/years | 310 | 46.20 ± 5.458 | | 46.21 ± 5.469 | | 0.010 | 0.992 |
| Shift work | | | | | | 0.000 | 1.000 |
| Yes | 216 | 108 | 50.0 | 108 | 50.0 | | |
| No | 94 | 47 | 50.0 | 47 | 50.0 | | |
| Ethnic group | | | | | | 1.741 | 0.187 |
| Han | 253 | 122 | 48.2 | 131 | 51.8 | | |
| Others | 57 | 33 | 57.9 | 24 | 42.1 | | |
| Educational level | | | | | | 3.780 | 0.052 |
| High school or below | 173 | 95 | 549 | 78 | 45.1 | | |
| College or above | 137 | 60 | 43.8 | 77 | 56.2 | | |
| Professional title | | | | | | 1.229 | 0.268 |
| Junior or below | 95 | 43 | 45.3 | 52 | 54.7 | | |
| Intermediate or above | 215 | 112 | 52.1 | 103 | 47.9 | | |
| Marital status | | | | | | 1.303 | 0.521 |
| Single | 13 | 8 | 61.5 | 5 | 38.5 | | |
| Married | 259 | 126 | 48.6 | 133 | 51.4 | | |
| Divorced/others | 38 | 21 | 55.3 | 17 | 44.7 | | |
| Personal income per month/Yuan | | | | | | 5.684 | 0.017 |
| <5,000 | 202 | 111 | 55.0 | 91 | 945.0 | | |
| ≥5,000 | 108 | 44 | 40.7 | 64 | 59.3 | | |
| Smoking | | | | | | 27.01 | <0.001 |
| Yes | 183 | 114 | 62.3 | 69 | 37.7 | | |
| No | 127 | 41 | 32.3 | 86 | 67.7 | | |
| Drinking alcohol | | | | | | 8.903 | 0.003 |
| Yes | 218 | 121 | 55.5 | 97 | 44.5 | | |
| No | 92 | 34 | 37.0 | 58 | 63.0 | | |
| BMI (Body mass index/kg·m$^{-2}$) | | | | | | 5.845 | 0.016 |
| 18.5~24 | 102 | 41 | 40.2 | 61 | 59.8 | | |
| >24 | 208 | 114 | 54.8 | 94 | 45.2 | | |
| Working age/years | | | | | | 1.685 | 0.194 |
| ≤15 | 113 | 51 | 45.1 | 62 | 54.9 | | |
| >15 | 197 | 104 | 52.8 | 93 | 47.2 | | |
| Total | 310 | 155 | 50.0 | 155 | 50.0 | | |

additive model (CC vs AA, OR = 1.913, 95% CI [1.085–3.375]) and the dominant model (AC/CC vs AA, OR = 2.012, 95% CI [1.200–3.373]), which could increase hypertension risk ($P < 0.05$).

**Table 3 Distribution of gene and occupational stress between case group and control group.**

| Item | N | Case group | | Control group | | $\chi^2$ | P |
|---|---|---|---|---|---|---|---|
| | | n | % | n | % | | |
| *SELE* A561C | | | | | | 1.025 | 0.599 |
| AA | 213 | 103 | 48.4 | 110 | 51.6 | | |
| CC | 15 | 9 | 60.0 | 6 | 40.0 | | |
| AC | 82 | 43 | 52.4 | 39 | 47.6 | | |
| *SELE* G98T | | | | | | 6.776 | 0.034 |
| GG | 196 | 87 | 44.4 | 109 | 55.6 | | |
| TT | 16 | 10 | 62.5 | 6 | 37.5 | | |
| GT | 98 | 58 | 59.2 | 40 | 40.8 | | |
| *MTHFR* A1298C | | | | | | 12.036 | 0.002 |
| AA | 102 | 37 | 36.3 | 65 | 63.7 | | |
| CC | 66 | 40 | 60.6 | 26 | 39.4 | | |
| AC | 142 | 78 | 54.9 | 64 | 45.1 | | |
| *MTHFR* C677T | | | | | | 7.130 | 0.028 |
| CC | 116 | 50 | 43.1 | 66 | 56.9 | | |
| TT | 52 | 34 | 65.3 | 18 | 34.6 | | |
| CT | 142 | 71 | 50.0 | 71 | 50.0 | | |
| Occupational stress | | | | | | 11.921 | 0.003 |
| Low | 23 | 8 | 34.8 | 15 | 65.2 | | |
| Middle | 127 | 52 | 40.9 | 75 | 59.1 | | |
| High | 160 | 95 | 59.4 | 65 | 40.6 | | |

## Effect of gene-gene and gene-occupational tension interactions on hypertension

GMDR software was used to analyze the effects of gene-gene and gene-occupational tension interactions on hypertension. The gene-gene interaction showed the best interaction model between the MTHFR gene A1298C and C677T loci with a training set precision of 0.6677 and a test set precision of 0.6639, a sign test $P = 0.001$, and a cross-validation agreement coefficient of 10/10, as shown in Table 5 and Fig. 1. The dendrogram showed a strong positive interaction between the MTHFR gene A1298C polymorphism and the C677T loci, see Fig. 2. The gene-occupational tension interaction showed the best interaction model between MTHFR gene A1298C polymorphism, C677T polymorphism and occupational tension with a training set precision of 0.7662, test set precision of 0.7440, sign test $P = 0.001$, and cross-validation consistency coefficient of 10/10, see Fig. 3. The dendrogram showed a strong positive interaction between the C677T polymorphism of the MTHFR gene and occupational tension, see Fig. 4.

## DISCUSSION

Hypertension is a major risk factor for cardiovascular disease and has a low control rate due to its complex and diverse etiology (*Gheorghe et al., 2018*). Many researchers have

**Table 4 logistic regression analysis of the influence of genes and occupational stress on the prevalence of hypertension.**

| Gene | Unadjusted | | Adjusted | |
|---|---|---|---|---|
| | OR (95% CI) | P | OR (95% CI) | P |
| SELE A561C | | | | |
| AA | 1.00 | | 1.00 | |
| AC | 1.073 [0.624–1.844] | 0.800 | 1.161 [0.654–2.058] | 0.610 |
| CC | 1.519 [0.496–4.652] | 0.465 | 1.667 [0.495–5.617] | 0.409 |
| Additive | 1.751 [0.547–5.601] | 0.345 | 1.602 [0.551–4.658] | 0.387 |
| Dominant | 1.243 [0.763–1.997] | 0.392 | 1.337 [0.800–2.235] | 0.268 |
| Recessive | 0.653 [0.227–1.882] | 0.430 | 0.603 [0.192–1.896] | 0.387 |
| SELE G98T | | | | |
| GG | 1.00 | | 1.00 | |
| GT | 2.039 [1.211–3.433] | 0.007 | 2.151 [1.227–3.375] | 0.008 |
| TT | 2.756 [0.924–8.216] | 0.069 | 2.724 [0.879–8.439] | 0.082 |
| Additive | 2.088 [0.730–5.971] | 0.170 | 2.086 [0.666–6.535] | 0.207 |
| Dominant | 1.852 [1.159–2.959] | 0.010 | 1.925 [1.163–3.186] | 0.011 |
| Recessive | 0.584 [0.207–1.648] | 0.309 | 0.602 [0.199–1.822] | 0.369 |
| MTHFR A1298C | | | | |
| AA | 1.00 | | 1.00 | |
| AC | 2.223 [1.279–3.864] | 0.005 | 1.917 [1.064–3.453] | 0.030 |
| CC | 2.682 [1.351–5.327] | 0.005 | 2.233 [1.082–4.609] | 0.030 |
| Additive | 2.703 [1.428–5.114] | 0.002 | 2.497 [1.277–4.883] | 0.007 |
| Dominant | 2.303 [1.414–3.752] | 0.001 | 2.012 [1.200–3.373] | 0.008 |
| Recessive | 0.579 [0.333–1.008] | 0.054 | 0.610 [0.337–1.104] | 0.103 |
| MTHFR C677T | | | | |
| CC | 1.00 | | 1.00 | |
| CT | 1.683 [0.990–2.860] | 0.054 | 1.913 [1.085–3.375] | 0.025 |
| TT | 2.511 [1.230–5.125] | 0.011 | 3.117 [1.430–6.795] | 0.004 |
| Additive | 2.493 [1.264–4.918] | 0.008 | 2.936 [1.398–6.166] | 0.004 |
| Dominant | 1.557 [0.980–2.575] | 0.061 | 1.802 [1.094–2.968] | 0.021 |
| Recessive | 0.468 [0.251–0.870] | 0.017 | 0.437 [0.225–0.849] | 0.015 |
| Occupational stress | | | | |
| Low | 1.00 | | 1.00 | |
| Middle | 1.300 [0.514–3.289] | 0.580 | 0.753 [0.277–2.051] | 0.579 |
| High | 2.740 [1.098–6.837] | 0.031 | 1.629 [0.610–4.352] | 0.330 |

done numerous studies on environmental and genetic aspects and found that occupational stress is one of the risk factors for hypertension, but MTHFR gene and SELE base polymorphisms may lead to different levels of individual susceptibility to hypertension depending on geographical regions and ethnicity. Previous studies have also suggested the utility of hypertension susceptibility genotypes in Mendelian randomization (*Fu et al., 2019*). However, it has been argued that a single genotype in the MTHFR gene and SELE

**Table 5 GMDR model of the influence of gene-gene, gene-occupational stress interaction on the prevalence of hypertension.**

| Interaction model | Accuracy | | Sign test (P) | CV consistency |
|---|---|---|---|---|
| | Training set | Testing set | | |
| Gene-Gene | | | | |
| A1298C | 0.5903 | 0.5851 | 9 (0.0107) | 10/10 |
| A1298C*C677T | 0.6677 | 0.6639 | 10 (0.0010) | 10/10 |
| A1298C*C677T*A561C | 0.6929 | 0.6542 | 10 (0.0010) | 7/10 |
| A1298C*C677T*A561C*G98T | 0.7310 | 0.6340 | 10 (0.0010) | 10/10 |
| Gene-Occupational stress | | | | |
| Occupational stress | 0.5993 | 0.5503 | 8 (0.0547) | 7/10 |
| Occupational stress*C677T | 0.7010 | 0.6586 | 10 (0.0010) | 9/10 |
| Occupational stress*C677T*A1298C | 0.7662 | 0.7440 | 10 (0.0010) | 10/10 |
| Occupational stress*C677T*A1298C*G98T | 0.7958 | 0.7351 | 10 (0.0010) | 10/10 |
| Occupational stress*C677T*A1298C*G98T*A561C | 0.8235 | 0.6916 | 10 (0.0010) | 10/10 |

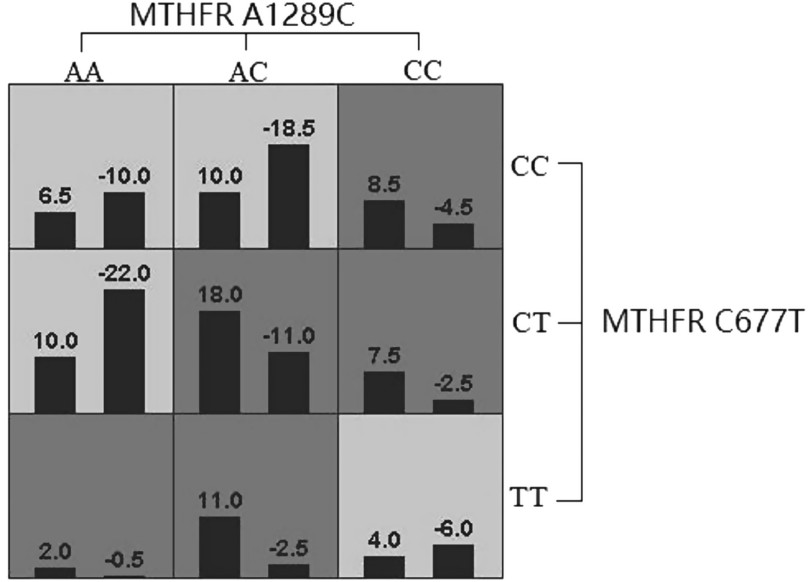

**Figure 1 Forest plot of logistic regression analysis of the influence of genes and occupational stress.**

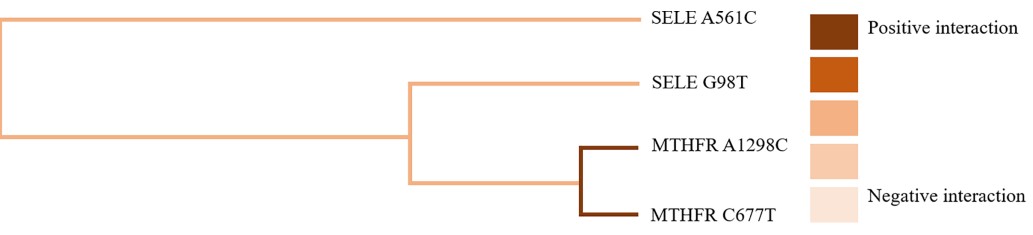

**Figure 2 Gene-gene interaction model of MTHFR gene A1298C and C677T sites.**

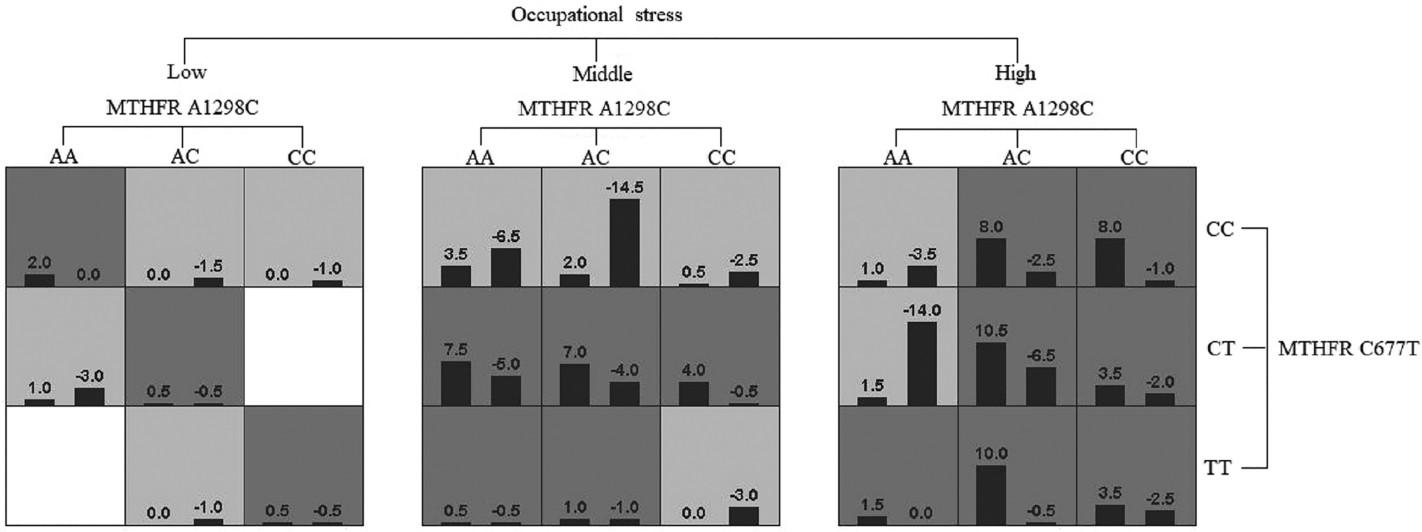

**Figure 3  Gene-gene interaction dendrogram of MTHFR gene A1298C and C677T sites.**   

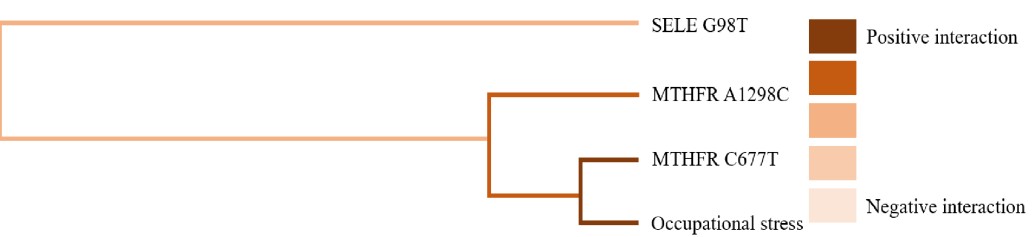

**Figure 4  Gene-environment interaction model of MTHFR gene A1298C and C677T sites and occupational stress.**   

gene should not be used as independent markers in a Mendelian randomization design, but rather the study should be expanded to include gene-environment interactions. Therefore, this study not only explored the effect of genes, but also the effect of gene-environment interactions on hypertension.

Studies about some populations in the three northeastern provinces of China have found that polymorphisms in the MTHFR genes A1298C and C677T are not significantly associated with hypertension (*Liu et al., 2019*). In this case-control study of oil workers in Karamay, Xinjiang, adjusting for confounding factors, mutant genes at the A1298C polymorphism and the C677T polymorphism of the MTHFR gene were found to be susceptible genotypes for hypertension, which is consistent with the findings of and *Alghasham et al. (2012)*. The human MTHFR gene is located on chromosome 1 (1P36.3) and has 11 exons, and the SNP sites that have a greater impact on enzyme activity are found at the C677T and A1298C loci, which are in the coding region.

The C677T polymorphism is a point mutation at the position 677 on MTHFR gene with the substitution of cysteine to thymine nucleotide at that position. This point mutation causes the substitution of alanine to valine in the MTHFR enzyme. The single nucleotide polymorphism of this gene reduces the thermostability of the MTHFR enzyme due to

the decreased activity of the enzyme at 37 °C or higher. MTHFR enzyme activity in homozygous subjects is 50–60% lower at 37 °C and 65% lower at 46 °C compared with normal non-mutated controls (*Rosenberg et al., 2019*). The inability of the MTHFR enzyme to catalyse the conversion of 5,10-methylenetetrahydrofolate to 5-methyltetrahydrofolate leads to the rise of plasma homocysteine levels in the homozygous mutated subjects. Some researchers postulated that homocysteine could cause atherogenesis and thrombogenesis leading to substantialfibrosis and muscle cell hyperplasia,which in turn can be a risk factor for coronary artery disease (*Stanger et al., 2004*). In addition, The MTHFR C677T polymorphism was also reported to be associated with increased risk of myocardial infarction in young/middle-aged Caucasians. Individuals with the MTHFR 677TT genotype and a low folate status had a significantly higher risk of coronary heart disease. A recent study also reported the association between MTHFR C677T gene polymorphism and essential hypertension which is closely related to the increased level of Hcy (*Ren, He & Cao, 2018*).

The present study also yielded such results. Numerous studies have confirmed that homocysteine (HCY) is an independent risk factor for cardiovascular disease and can cause vascular lesions in the body, and HCY accumulation caused by MTHFR mutations may increase the risk of vascular complications in hypertensive patients. Scholars have demonstrated that HCY levels are significantly elevated in individuals carrying the 677TT genotype, suggesting that the mechanism of MTHFR C677T and the development of hypertension maybe related to abnormal HCY metabolism due to decreased mutant enzyme activity, which impairs the normal biological function of endothelial cells, thereby inducing oxidative stress that exacerbates the inflammatory response and puts the blood vessels in a persistent inflammatory state, etc. It has been suggested that the A1298C mutation does not inherently affect blood pressure levels but may play a regulatory role when individuals are accompanied by the MTHFR C677T mutation. In contrast, the present study showed that the mutant gene at the A1298C polymorphism of the MTHFR gene is a susceptible genotype for hypertension, which may be due to different geographical and ethnic differences.

When the A-C mutation occurs in the A1298C polymorphism, the transcriptionally expressed glutamate will be converted to alanine, which will affect the catalytic regulation of the enzyme. A case-control study found the A1298 C polymorphisms were related to depression severity (*Chen & Chang, 2021*). It has been suggested that although the A1298C mutation itself does not affect blood pressure levels, it may play a regulatory role when individuals are accompanied by the MTHFR C677T mutation. In contrast, the present study showed that the mutated gene with the A1298C polymorphism in the MTHFR gene is a susceptible genotype for hypertension, which may be due to differences across geography and ethnicity. The difference in the association of C677T and A1298C with hypertension may be due to their gene polymorphism, with the C677T mutation site present in the MTHFR gene The C677T mutation site is found in exon 4 of the MTHFR gene and is involved in encoding the catalytic structural domain of the N-terminal part of the enzyme, which directly affects the catalytic activity of the enzyme, whereas the

A1298C mutation site is found in exon 7 of the MTHFR gene and encodes the C-terminal regulatory domain.

The study also found that the mutant gene at the G98T polymorphism of the SELE gene was the susceptible genotype for hypertension, which is consistent with the study of *Yesheng Wei et al. (2003)*. The human SELE polymorphism, located on the long arm of chromosome 1, is a 13-kb DNA sequence that is a vascular endothelial adhesion molecule and is a marker of vascular endothelial function. Activation of the endothelium promotes atherosclerosis reduces the elasticity of the arterial wall and alters the responsiveness of the endothelium to vascular stimuli, and endothelial dysfunction maybe a major cause of hypertension. The G98T polymorphism is a single nucleotide polymorphism in the coding region caused by the mutation of G to T in the 5′ untranslated region of exon 2 of the E-selectin gene at the site 98 (98bp downstream of the transcription start site). It was found that T allele carriers are at high risk of hypertension and have high blood pressure values. The G98T polymorphism mutation can enhance the expressed SELE function by affecting the E-selectin 5' untranslated region structure, thus causing enhanced adhesion, which in turn causes an inflammatory response, damage to endothelial cells, inhibition of NO and prostaglandin production, and diminished vascular responsiveness to endothelium-dependent vasotransfer substances, resulting in increased vascular resistance and, consequently, hypertension (*Srivastava et al., 2018*). There are fewer studies on SELE gene polymorphisms in other diseases. The SELE gene also appears to be associated with atherosclerosis, with monocytes adhering to vascular endothelial cells and migrating into the endothelium to take up lipids and transform into foam cells as an early event in the formation of atherosclerosis. Early events in the formation of atherosclerosis. Whereas there is no association between A561C gene polymorphism and hypertension. The differences between these studies may be due to epigenetic mechanisms involved in gene expression influenced by environmental conditions (*e.g.*, lifestyle and diet).

A meta-study in a population of pregnant women found that the risk of hypertension was significantly increased in the presence of high air pollution and mutations in the MTHFR gene (*Yang, Yang & Shiao, 2018*), and a study by *Fu et al. (2018)* in Chinese children confirmed that there was also an interaction between the MTHFR gene and the effect of obesity on hypertension. *Wu et al. (2015)* showed that the interaction between internal environmental factors may be a potential factor contributing to elevated blood pressure. In this study, we used GMDR software to construct risk models for the effects of gene-gene and gene-environment associations on hypertension, and further analysis of the interactions between occupational stress and the MTHFR and SELE genes found strong positive between the A1298C polymorphism of the MTHFR gene and the C677T polymorphism, and between the C677T polymorphism of the MTHFR gene and occupational stress. *Taylor et al. (2010)* did interaction validation of different genes with the environment and came to similar conclusions. The influence of genes on the pathogenesis of hypertension may be the increased deposition of extracellular matrix components and their altered structure or cell-extracellular matrix attachment, which causes structural changes in the arterial wall. In contrast, the MTHFR gene is an important

enzyme for folate metabolism, and mutations in the MTHFR gene cause hyperhomocysteinemia and homocystinuria, indirectly affecting changes in human blood pressure (*Bayramoglu et al., 2015*). SELE gene, on the other hand, plays a key role in the binding of lymphocytes and monocytes to endothelial cells. Its genetic polymorphisms may upregulate gene expression levels and thus affect the biological functions of its proteins. Several lines of evidence have shown that A561C and G98T gene polymorphisms can lead to a significant increase in blood pressure (*Faruque et al., 2011*). The special nature of oilfield operations, which often involve shift work, job evaluation, and learning new technologies lead to the occurrence of occupational stress as oilfield workers are often faced with loneliness, insomnia, and depression separated from their families (*Yong et al., 2020*). When both genetic and environmental factors are present, the risk of developing hypertension increases significantly.

## CONCLUSIONS

In conclusion, the risk of hypertension among oil workers in Karamay, Xinjiang, maybe influenced not only by environmental but also by genetic factors, and there is a strong interaction between the two, which provides an updated vision for the prevention and control of hypertension. This study demonstrated a strong interaction between gene-gene and gene-environment effects on hypertension, but there are several potential limitations that should not be overlooked: first, the sample size was small and there may be selection bias. Second, the genotyping method used in this study may have potential bias, and there are reports demonstrating higher sensitivity and accuracy of genotyping methods such as Mass Array and gene chips. Therefore, further sample expansion and selection of more appropriate genotyping methods must be performed in the future to confirm the results we observed in this study.

## ACKNOWLEDGEMENTS

The authors thank all participants and investigators.

### Funding

This work was supported byXinjiang Uygur Autonomous Region Natural Science Foundation (2020D01C158; 2018D01C167; 2016D01C173). The funders had no role in study design, data collection and analysis, decision to publish, or preparation of the manuscript.

### Grant Disclosures

The following grant information was disclosed by the authors:
Xinjiang Uygur Autonomous Region Natural Science Foundation: 2020D01C158; 2018D01C167; 2016D01C173.

### Competing Interests

The authors declare that they have no competing interests.

## PeerJ

## Author Contributions

- Fen Yang conceived and designed the experiments, performed the experiments, analyzed the data, prepared figures and/or tables, authored or reviewed drafts of the paper, and approved the final draft.
- Ruiying Qiu performed the experiments, analyzed the data, prepared figures and/or tables, and approved the final draft.
- Saimaitikari Abudoubari performed the experiments, prepared figures and/or tables, and approved the final draft.
- Ning Tao conceived and designed the experiments, authored or reviewed drafts of the paper, and approved the final draft.
- Hengqing An conceived and designed the experiments, analyzed the data, authored or reviewed drafts of the paper, and approved the final draft.

## Human Ethics

The following information was supplied relating to ethical approvals (*i.e.*, approving body and any reference numbers):

The study was approved by the Ethics Review Committee of the First Affiliated Hospital of Xinjiang Medical University (No. 2015006).

## Data Availability

The raw measurements are available in the Supplemental Files.

## Supplemental Information

Supplemental information for this article can be found online at http://dx.doi.org/10.7717/peerj.12914#supplemental-information.

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
