# Peer review of "Effect of interaction between occupational stress and polymorphisms of MTHFR gene and SELE gene on hypertension"

_PeerJ, doi:10.7717/peerj.12914_

## Round 0.1 · original submission · Major Revisions

The reviewers have found scientific merit in your work, although there are some issues which you should address in a revised version of the text.

·

Basic reporting

No comments

Experimental design

1.- Is not clear what methods were used for the determination of the polymorphisms. The authors have only shown the primers for the amplification and the PCR conditions, but how was detected the specific allele?. This point must be clarified.

2.- In my opinion, tables 1-3 are not necessary for the manuscript. Some of this information can be included in the text, in the section on material and methods.

3.- I think that is better the term “polymorphism” instead than “locus”.

Validity of the findings

1.- Table 5 shows the distribution of the polymorphisms and occupational stress in cases and controls. One logistic regression analysis under inheritance genetic models (recessive, additive, and dominant) in order to establish the possible association of the polymorphisms with risk to present hypertension is necessary before the interaction analysis.

2.- Do the polymorphisms studied have some functional effect? Where are they located, in promoter or coding regions? This point should be discussed.

Additional comments

In the present work, the authors determined two MTHFR and two SELE polymorphisms in patients with and without hypertension in order to establish the interaction of these polymorphisms with occupational stress. The study included 155 individuals with hypertension and 155 without this condition. The results suggest a strong interaction between gene-gene and gene-environment on hypertension. This is an interesting study with some limitations that are commented on by the authors. Some points should be clarified.

Reviewer 2 ·

Basic reporting

1. Table 4 percentages: Should be calculated for different variables within each group, rather than between groups. And t-tests were used to compare the ages of the two groups.
2. Table 5 gene percentages: The percentage of each genotype within different groups should be calculated and additionally OR values could be included to make the results clearer. The genotypes should also preferably be in the order of wild-type pure, heterozygous and finally mutant pure.

Experimental design

Lack of description of the process of sample collection and DNA extraction.

Validity of the findings

1. "The genotypes GT at locus G98T of the SELE gene, AC and CC at locus A1298C of the MTHFR gene, and CT and TT at locus C677T of the MTHFR gene were associated with the prevalence of hypertension with OR (95% CI) of, respectively, GT: 2.151 (1.227-3.375), AC : 1.917 (1.064-3.453), CC: 2.233 (1.082-4.609), CT: 1.913 (1.085-3.375), TT: 3.117 (1.430-6.795)" This presentation is rather confusing and it is recommended to refer to other literature for a more scientific presentation.
2.It is suggested that some conclusions of the manuscript should not be expressed in a more positive way.

Additional comments

This study was conducted in Xinjiang oil workers to investigate the effects of occupational stress and genetic polymorphisms in the MTHFR and SELE genes on the development of hypertension in occupational personnel. It has strong relevance for the prevention and treatment of hypertension in occupational populations. The paper is good, but should improve some aspects of the paper.

---

## Round 0.2 · Minor Revisions

Still pending some minor changes:
1. "Locus" is still in the manuscript.
2. Including probability in the conclusion.

---

## Round 0.3 · Major Revisions

Please, address the comments indicated by the reviewer in a point-by-point letter and showing your changes in the revised manuscript.

·

Basic reporting

No comments

Experimental design

1.- Is not clear what methods were used for the determination of the polymorphisms. The authors have only shown the primers for the amplification and the PCR conditions, but how was detected the specific allele?. The authors used the PCR-RFLP. What enzyme was used and what fragments were obtained for each allele/genotype?. This point must be clarified.

2.- Table 5 shows the distribution of the polymorphisms and occupational stress in cases and controls. One logistic regression analysis under inheritance genetic models (recessive, additive, and dominant) in order to establish the possible association of the polymorphisms with risk to present hypertension is necessary before the interaction analysis.

Validity of the findings

No comments

Additional comments

1.- Do the polymorphisms studied have some functional effect? Where are they located, in promoter or coding regions? This point should be discussed.

---

## Round 0.4 · accepted · Accept

No more comments after this version of the text.

·

Basic reporting

No comments

Experimental design

No comments

Validity of the findings

No comments

Additional comments

No comments